# The Influence of Fresh Latex Coagulation on the Parameter Characteristics of the Yeoh Hyperelastic Constitutive Model for Natural Rubber

**DOI:** 10.3390/polym16243601

**Published:** 2024-12-23

**Authors:** Li Ding, Honghai Huang, Yuekun Wang, Jianwei Li, Hongxing Gui, Yongping Chen

**Affiliations:** 1School of Materials Science and Engineering, Hainan University, Haikou 570228, China; gdzjding@163.com; 2Rubber Research Institute, Chinese Academy of Tropical Agricultural Science, Haikou 570101, China; honghaihuang2009@163.com (H.H.); xwangyk@126.com (Y.W.); 18272733025@163.com (J.L.); 3Hainan Natural Rubber Technology Innovation Center, Haikou 571101, China

**Keywords:** natural rubber (NR), coagulation, constitutive model, Yeoh model, hyperelastic

## Abstract

The coagulation of fresh latex is one of the critical processes that impacts rubber quality during natural rubber processing. Constitutive relationships are the basis for the study of the mechanical properties of rubber materials and serve as a prerequisite for material simulation studies. However, studies on the effect of different coagulation methods on natural rubber constitutive relationships have yet to be carried out, and the current models used for natural rubber constitutive relationships need to be improved. In order to investigate the effects of different coagulation methods on the hyperelastic properties of natural rubber, the impact of natural coagulation, enzyme coagulation, acid coagulation, microbial coagulation, and enzyme-assisted microbial coagulation on the hyperelastic constitutive relationship of natural rubber were analyzed in detail based on tensile experiments and the Yeoh model. The results show that after introducing a strain rate-related factor, the Yeoh model can describe well the mechanical behavior of natural rubber carbon black composites in different deformation regions, and the rubber, studied with varying coagulation methods, exhibits different mechanical properties in different deformation regions. This study provides new evidence for the study of high-performance natural rubber and serves as a reference for process selection in the primary processing of natural rubber.

## 1. Introduction

Natural rubber is a primary processed product obtained from the fresh latex of the Brazilian rubber tree through processes such as coagulation, dehydration (sheet and crepe processing), and drying. The structural composition of natural rubber and other bioactive components endow it with superior resilience, insulation, waterproofness [1], and flexibility compared to synthetic rubber. As a result, natural rubber is extensively applied in the tire manufacturing industry and remains irreplaceable in some critical industrial sectors and cutting-edge equipment. With the continuous advancement of science and technology and the expansion of various fields, higher demands have been placed on the comprehensive performance of natural rubber.

The coagulation of fresh latex is a critical step in the initial processing of natural rubber that significantly affects rubber quality [2], primarily because non-rubber substances such as proteins and phospholipids are key components of the natural rubber network structure and are closely related to the vulcanization characteristics and mechanical properties of natural rubber. The types and content of these non-rubber components can change variably under different coagulation processes, thus causing variations in the quality and application performance of natural rubber [3]. The common coagulation processes and their characteristics are as follows: Acid coagulation is the most widely used method but can corrode equipment and affect product performance [4]. Natural coagulation requires no operation but is time-consuming and may result in incomplete coagulation. It is the method used for obtaining cup lump raw materials [5,6]. Microbial coagulation produces dry rubber with better physical–mechanical properties and vulcanization characteristics [7], but wet coagulated blocks can develop an odor during storage, similar to naturally coagulated blocks [8,9]. Enzymatic coagulation has a rapid coagulation rate and high product physical–mechanical properties but leads to fast vulcanization and the risk of scorching the rubber compound. Enzyme-assisted microbial coagulation combines the advantages of microbial and enzymatic coagulation, improving vulcanization processing characteristics while obtaining higher mechanical properties [10].

Choosing the appropriate coagulation process is an important method to improve the performance of rubber products. Currently, evaluations of fresh latex coagulation processes mainly focus on the effects of latex coagulation and specific performance indicators of raw rubber and vulcanized rubber [11,12,13]. Ng et al. [2] described the subsequent effects of different coagulation methods on natural rubber from the perspective of processing characteristics, vulcanization characteristics, mechanical properties, and the aging retention performance of vulcanized rubber. However, to establish a connection between latex coagulation technology and the rubber industry, it is necessary to research the impact of different coagulation methods on the application performance of end products. Forming and structural simulations are common methods for predicting the performance of rubber composites and the application performance of end products due to their convenience, speed, and intuitiveness [14,15,16,17,18]. Engineers often use software such as Ansys and Abaqus for simulations [19,20,21,22], and the constitutive model and parameters play an important role in the reasonableness and precision of the calculations.

There are few references, mainly in Chinese, on using constitutive models to study the hyperelastic properties of natural rubber. The Mooney–Rivlin [23,24,25] and Yeoh [26,27] models are commonly used to investigate the constitutive relationships of natural rubber. Zhou et al. [28] found that the Mooney–Rivlin model is suitable for small to medium deformations, generally for strains of about 100% for tension and 30% for compression, and gives relatively accurate results for rubber without carbon black. However, it fails to simulate the behavior of carbon black-filled rubber composites precisely. Zhao et al. also found that the Mooney–Rivlin model cannot accurately reflect the stress–strain relationship of hyperelastic material under large deformation, and the fitting degree of uniaxial tensile data of two diaphragm materials is relatively low [29]. The Yeoh model has the advantage of simulating mechanical behavior under other deformation conditions using simple uniaxial tensile test data [30]. Wang found that the Yeoh model is better suited to simulating the significant deformation behavior of carbon black-filled natural rubber [31]. However, the Yeoh model cannot accurately simulate the mechanical behavior of small deformation regions. Huang et al. compared the displacement and stress maps of the two models with the Ansys finite element analysis software. They found that the Mooney–Rivlin model is suitable for simulating medium and small deformation behaviors, and the Yeoh model is more suitable for simulating the significant deformation behavior of carbon black-filled natural rubber [32]. Carbon black is the most commonly used filler in the natural rubber industry. The Yeoh model cannot accurately simulate the small deformation region primarily because the initial tensile speed must rapidly increase from zero to the set speed within a short period, resulting in a non-constant strain rate in the small strain area, and thus, inaccurate simulation results [28].

In order to investigate the effects of different coagulation methods on the hyperelastic properties of natural rubber, this study applies the Yeoh model in conjunction with a tensile testing machine to explore the constitutive relationships of carbon black-filled natural rubber (NR) samples obtained through various coagulation methods. It examines the variations in Yeoh model parameters and the mechanical properties of natural rubber under different coagulation techniques. Furthermore, a strain-rate-related factor is introduced to refine the Yeoh model, addressing its limitation in accurately describing the mechanical behavior of natural rubber carbon black composites in the low-strain region. This article lays the foundation for further research on material forming and structural simulation.

## 2. Materials and Methods

### 2.1. Raw Materials and Reagents

Fresh latex was collected from the Chinese Academy of Tropical Agricultural Sciences experimental field in Danzhou, Hainan, China. The microbial coagulation liquid was prepared and preserved by the Rubber Research Institute of the Chinese Academy of Tropical Agricultural Sciences. The alkaline protease was sourced from Shengwanjia New Materials Co., Ltd., Ji’nan, China. The other chemical reagents were commercially available.

### 2.2. Instruments and Equipment

The instruments and equipment were sourced from the following locations: the electronic tensile machine: AI-7000-SGD1, Gotech Testing Machines Inc., Qingdao, China. the open mill: XK-300B, Shanghai Kechuang Rubber and Plastic Machinery Co., Ltd., Shanghai, China. the rheometer: MDR-2000E, Liyuan Electronic Chemical Equipment Co., Ltd., Dongguan, China. and the flat vulcanizing machine: JH-PB-600T, Shanghai Kuntian Laboratory Instrument Co., Ltd., Shanghai, China.

### 2.3. Sample Preparation

Homogeneous ammonia-free fresh latex with a dry rubber content of 23% was taken and sampled using different coagulation methods, which are shown in Table 1.

(1)Natural coagulation: No additives were used, allowing the latex to coagulate by standing still.(2)Enzymatic coagulation: A protease solution was added to the latex, mixed well, and then left to coagulate by standing still. The amount of protease used was 0.09% of the latex’s dry rubber content.(3)Acid coagulation: A formic acid solution was added to the latex, mixed well, and then left to coagulate by standing still. The amount of formic acid used was 0.35% of the latex’s dry rubber content.(4)Microbial coagulation: Microbial coagulation liquid was added to the latex, mixed well, and left to coagulate by standing still. The microbial coagulation liquid used was 10% of the latex weight.(5)Enzyme-assisted microbial coagulation: The protease solution was first added to the latex, mixed well, and left to stand for 20 min. Then, the microbial coagulation liquid was added, mixed well, and left to coagulate by standing still. The amount of protease used was 0.05% of the latex’s dry rubber content, and the amount of microbial coagulation liquid used was 10% of the latex weight.

After the above latex coagulated and matured for 2 days, it was pressed into sheets, hung to dry in a ventilated area for 3 days, and dried at 80 °C to produce raw rubber samples prepared by different coagulation methods.

### 2.4. Preparation of NR Vulcanizates

Compounded rubber was prepared as shown in Table 2. Firstly, the rubber sample was loaded into an open mill and plasticized for 4 min. Then, the vulcanization ingredient (ZnO, Stearic acid, sulfur, TBBS, Oil furnace Black) was added to the sample and mixed for another 7 min at 67 °C. Finally, the curing properties of the rubber compounds were characterized with a rheometer, and the vulcanization of the NR sheets was completed at 143 °C for the optimum cure time (tc90) by a flat vulcanizing machine.

### 2.5. Sample Testing

The dumbbell-shaped vulcanizates were prepared as shown in Figure 1a.

(1)Uniaxial tensile test: The electronic tensile machine obtained the tensile properties and stress–strain curves, shown in Figure 1b.(2)Cyclic tensile test: A tensile strain of 400% was selected for five load–unload cycles at a stretch rate of 500 mm/min by the electronic tensile machine.

### 2.6. Yeoh Model Under Uniaxial Tensile Conditions for Rubber [31]

The Yeoh model is a commonly used molecular statistical constitutive model for rubber materials, which is particularly suitable for simulating the large deformation behavior of carbon black-filled natural rubber. The ANSYS finite element analysis software also categorizes it into various parameter forms such as first, second, third, fourth, and fifth orders. Rubber, being an incompressible material, has its strain energy density function model under uniaxial tensile conditions with three parameters, as follows [31]:(1)$$W=\sum_{i=1}^{3}C_{i0}{\left(I_{1}-3\right)}^{i}$$
where W is the strain energy function, *I*_1_ is the first invariant of the deformation tensor, and C*_i_*_0_ is a material constant determined by material experimentation, with the initial shear modulus μ = 2C_10_. For uniaxial tension and incompressible materials [33] the following equation applies:(2)$$I_{1}={\lambda_{1}}^{2}+\frac{2}{\lambda_{1}}$$

Because [34]
(3)$$\lambda_{1}=1+\frac{\Delta L}{L_{0}}=1+\varepsilon$$

Then,
(4)$$I_{1}={\left(1+\varepsilon \right)}^{2}+2{\left(1+\varepsilon \right)}^{-1}$$
where $\lambda_{1}$ is the elongation, $L_{0}$ is the initial length, ΔL is the deformation length, and ε represents the engineering strain. The three-parameter Yeoh constitutive relationship model for rubber is as follows [31]:(5)$$\;\sigma =2\left[\left(1+\varepsilon \right)-(1+\varepsilon {)}^{-2}\right]\left\{C_{10}+2C_{20}\left[{\left(1+\varepsilon \right)}^{2}+2{\left(1+\varepsilon \right)}^{-1}-3\right]+3C_{30}{\left[{\left(1+\varepsilon \right)}^{2}+2{\left(1+\varepsilon \right)}^{-1}-3\right]}^{2}\right\}$$

By conducting uniaxial tensile tests on rubber samples prepared using different coagulation methods, the strain and stress data obtained from the tests were nonlinearly curve-fitted by the above equations in the software Origin 2021 to obtain the parameters C_10_, C_20_, and C_30_ of the Yeoh model.

### 2.7. Strain Rate-Related Three-Parameter Yeoh Model

The Yeoh model cannot well simulate the stress–strain curve in the small strain region, mainly because, at the initial stretch, the stretching speed must rapidly increase from zero to the specified rate within a short period, resulting in a non-constant strain rate in the small strain region. Since the Yeoh model is a function related to the principal strain obtained by fitting uniaxial tensile test data and does not contain parameters and variables related to strain rate and time, it causes the fitting curve in the small strain area (<100%) to not coincide with the experimental curve. To solve this problem, different strain rates are designed for uniaxial tensile tests in the small strain region to refit the model parameters, and a strain rate-related factor ט is introduced to modify the Yeoh model equation.

### 2.8. The Determination and Application of the Modified Yeoh Model Parameter μ [28]

A group of samples were stretched to 100% strain at different tensile speeds of 100 mm/min, 200 mm/min, 300 mm/min, 400 mm/min, and 500 mm/min to measure the stress–strain curve data at each tensile speed. Selecting a certain tensile speed as the reference speed, the strain rate-related parameter μ was linearly fitted based on the introduced strain rate-related factor relationship, combined with unidirectional tensile test data and parameters simulated by the Yeoh model.

The fitted strain rate-related parameter μ was substituted into the modified Yeoh model, which was used to predict the uniaxial tensile stress–strain curves of the group of rubber materials at different strain rates within the small strain range. The predictions were then compared with the actual stress–strain curves to verify the model’s accuracy.

## 3. Results and Discussion

### 3.1. Tensile Property and Stress–Strain Curves of Natural Rubber Materials Under Different Coagulation Methods

The tensile property of natural rubber materials by different coagulation methods is shown in Table 3. Using the common acid coagulation process in production as a reference group, we explored the relative significance of the differences between natural coagulation, enzymatic coagulation, microbial coagulation, and enzyme-assisted microbial coagulation with acid coagulation and found that the relative differences between the different coagulation modes were highly significant, which are shown in Table 4 and Table 5.

The stress–strain curves for rubber obtained through five different coagulation methods exhibit a consistent trend, which is shown in Figure 2. The differences become increasingly evident in the medium-to-large strain region, i.e., beyond 200%. Regarding mechanical properties, the samples produced by enzyme-assisted microbial coagulation exhibited higher elongation at break and slightly greater tensile strength. In contrast, the tensile strength of natural and acid coagulation was relatively lower. It is also noticeable that at the same strain, the modulus of the samples from enzymatic and microbial coagulation was relatively higher, which is primarily because non-rubber components facilitate the construction of the natural rubber vulcanization network [3]. Enzymatic and microbial coagulation involves more intense material transformations and a richer variety of non-rubber components, resulting in a more extensive network crosslinking and a better overall performance than rubber obtained through natural and acid coagulation [5,35]. Because enzymes can degrade non-rubber components such as proteins to produce small molecules [10], and the growth and metabolism of microorganisms can also convert these non-rubber components into small molecules [35], they have similar effects, and so the stress–strain curves of enzymatic coagulation and microbial coagulation rubber are close to each other. As for natural coagulation and the acid coagulation, the exogenous substances that promote the change in their non-rubber components are relatively few, so they also have similar stress–strain curves.

### 3.2. The Impact of Coagulation Method on the Yeoh Model Parameters Under the Uniaxial Tensile Conditions for Rubber

The stress–strain data from uniaxial tensile tests on rubber samples produced by different coagulation methods were fitted using the Yeoh model equation. Comparisons between the five fitted curves and the experimental stress–strain curves are shown in Figure 3, and the Yeoh model parameters C_10_, C_20_, and C_30_ were obtained, as shown in Table 6. These parameters directly affect the accuracy and effectiveness of the simulation results.

C_10_ represents the initial shear modulus. As the second coefficient, C_20,_ is positive, the material hardens during moderate deformation, but due to the third coefficient, C_30,_ being negative, it softens under large deformations. The small strain region has a larger error, but the results align well with experimental outcomes in the moderate to large strain regions.

Since the material parameter C_10_ represents half of the shear modulus at a small strain region [36], the initial shear modulus of the rubber samples prepared by the five coagulation methods, from largest to smallest, is as follows: natural coagulation, microbial coagulation, enzymatic coagulation, enzyme-assisted microbial coagulation, and acid coagulation. The material parameter C_20_ indicates the hardening phenomenon of filled rubber under moderate deformation. The greater C_20_, the harder the rubber material is [37]. It can be seen that in moderate deformation, the hardness of rubber samples prepared by the five coagulation methods, from hardest to softest, is as follows: enzymatic coagulation, microbial coagulation, acid coagulation, natural coagulation, and enzyme-assisted microbial coagulation. The material parameter C_30_ indicates the softening of filled rubber under large deformation. The larger C_30_, the more easily the rubber material softens [37]. Therefore, the order of the rubber samples prepared by the five coagulation methods in terms of ease of softening under large deformation is as follows: natural coagulation, enzymatic coagulation, acid coagulation, microbial coagulation, and enzyme-assisted microbial coagulation.

### 3.3. Effect of Coagulation Method on Energy Dissipation Capacity of Natural Rubber Composites During Cyclic Stretching

Observations from the cyclic stretching curves at a fixed strain of 400% and the energy dissipation of the five samples show that the energy dissipation decreases gradually with an increasing number of cycles, exhibiting the typical Mullins effect [38], which is shown in Figure 4. The differences in energy dissipation are more pronounced during the first stretch but become less significant thereafter, as shown in Figure 5. For rubber materials, a portion of the dissipated energy contributes to deforming the network structure, leading to material deformation and thus providing the rubber with shock-absorbing functionality. The extent of deformation is related to the softness or hardness of the material; hence, samples with greater energy dissipation indicate a tendency for easier deformation or relatively softer characteristics. Therefore, the results of the cyclic stretching tests suggest that the coagulation methods leading to easier deformation of the rubber material are, in order, natural coagulation, enzymatic coagulation, acid coagulation, microbial coagulation, and enzyme-assisted microbial coagulation—consistent with the hardness rankings under the large strain predicted by the Yeoh model.

### 3.4. Optimization of the Yeoh Model Parameters

Comparing the fitted curves of the Yeoh model with the experimental curves, the Yeoh model can describe the mechanical behavior of natural rubber carbon black composites well in the moderate and large strain regions. However, there is a significant error in the small strain region. It has been proposed that different forms of constitutive models must be adopted within different strain ranges [39], but this approach is cumbersome. To address this issue, uniaxial tensile tests with different strain rates were conducted in the small strain region to refit the model parameters, introducing a strain rate-related factor, טט, to modify the Yeoh model equation.

Based on previous research, the strain rate-related factor, ט, can be represented by the following exponential function [28]:(6)$$\upsilon =1+\mu \ln V_{\varepsilon }^{*}$$
(7)$$V_{\varepsilon }^{*}=\frac{V_{\varepsilon }^{f}}{V_{\varepsilon_{0}}}$$
where μ is the strain rate-related parameter, $V_{\varepsilon }^{*}$ is the ratio of strain rates, $V_{\varepsilon }^{f}$ is the actual strain rate obtained by the tensile machine at a certain stretching speed, and $V_{\varepsilon_{0}}$ is a reference strain rate taken from one of the rates in a set of experiments.

After introducing the strain rate-related factor, the Yeoh model can be written as follows:(8)$$\sigma =2\left[\left(1+\varepsilon \right)-(1+\varepsilon {)}^{-2}\right]\left\{C_{10}+2C_{20}\left[{\left(1+\varepsilon \right)}^{2}+2{\left(1+\varepsilon \right)}^{-1}-3\right]+3C_{30}{\left[{\left(1+\varepsilon \right)}^{2}+2{\left(1+\varepsilon \right)}^{-1}-3\right]}^{2}\right\}(1+\mu ln{V_{\varepsilon }}^{*})$$

This is the three-parameter Yeoh model related to the strain rate, and the modified Yeoh model can reflect the influence of the strain rate on the hyperelastic properties of the rubber material in the small strain region.

### 3.5. Determination of the Modified Yeoh Model Parameter μ

A rubber sample prepared by the natural coagulation method was selected and cut into several dumbbell-shaped films. The films were stretched to 100% strain at different stretching speeds, and the differences between the theoretical and actual strain rates at different speeds are shown in Table 7. The strain rate was calculated as the derivative of strain with respect to time because the scale distance of the tension machine was 25 mm, and the material was stretched to 100% with 100 mm/min, which is equivalent to four strains in 60 s, so the theoretical strain rate was 0.0667 s^−1^, but because the stretching speed needed to be increased from 0 to 100% mm/min, the actual stretching required a more extended period, leading to the actual strain rate being lower than the theoretical strain rate.

A stretching speed of 300 mm/min as the reference speed was selected, the actual strain rate at this speed was set as the reference strain rate $V_{\varepsilon_{0}}$, and the model parameter C_10_ at this speed was set as the reference $C_{10}^{\varepsilon_{0}}$; from this, the relationship between the strain rate $V_{\varepsilon }^{f}$, the model parameter $C_{10}^{f}$ at other stretching speeds and the reference speed $V_{\varepsilon_{0}}$, $C_{10}^{\varepsilon_{0}}$ could be calculated as follows [29]:(9)$$C_{10}^{f}=C_{10}^{\varepsilon_{0}}[1+\mu \ln\left(\frac{V_{\varepsilon }^{f}}{V_{\varepsilon_{0}}}\right)]$$

Equation (9) was transformed into:(10)$$\frac{C_{10}^{f}}{C_{10}^{\varepsilon_{0}}}-1=\mu \ln\left(\frac{V_{\varepsilon }^{f}}{V_{\varepsilon_{0}}}\right)$$

Combining the uniaxial tensile test data and the parameters simulated by the Yeoh model, the strain rate-related parameter μ could be linearly fitted using the equation above.

The actual stress–strain curve at a reference speed of 300 mm/min and the fitted curve of the Yeoh model are shown in Figure 6.

The reference strain rate $V_{\varepsilon_{0}}$ at the reference speed was 0.1105/s, and the simulated C10, which is $C_{10}^{\varepsilon_{0}}$, was 0.01273. The strain rates $V_{\varepsilon }^{f}$ and simulated parameters $C_{10}^{f}$ at other stretching speeds are shown in Table 8. According to Equation (10), the strain rate-related parameter μ can be linearly fitted to be 0.251, which is shown in Figure 7.

For the same batch of rubber samples, the strain rate-related parameter μ was constant. For different batches of samples, the strain rate-related parameter μ would not the same and would need to be recalculated according to the same method.

### 3.6. Application of the Modified Yeoh Rubber Model

The strain rate-related parameter (μ = 0.251) was substituted into the Yeoh-modified model (8) and the same sample was selected in 2.5 as the experimental object. Using the stress–strain curve data of the sample at a stretching speed of 300 mm/min as a basis, the theoretical strain data for each speed were calculated by substituting the values of C_10_, C_20_, and C_30_ at a 300 mm/min tensile speed and “ln($V_{\varepsilon }^{f}$/$V_{\varepsilon_{0}}$)” at this speed under a 300 mm/min tensile speed, as seen in Table 7 into Equation (8). The mechanical behavior at different strain rates was forecasted, with results depicted, which are shown in Figure 8. It can be observed that the predictions of the modified model at various stretching speeds align well with the actual stress–strain curves, especially at a stretching speed of 500 mm/min, where the stress–strain curve shows a marked improvement in the small strain region compared to the unmodified model.

The stress–strain data at different speeds predicted by the model and obtained from the actual test were fitted to a nonlinear curve in the software Origin 2021. The fitting function was Equation (8), which can be obtained as the goodness-of-fit R^2^, which is shown in Table 9. The higher the value of R^2^, the better the model fits the data, and R^2^ of the model prediction was 1, which indicates that it fits perfectly. In contrast, the actual test had an R^2^ of more than 0.99, which is close to the model prediction, indicating that the two curves have a high degree of approximation, simultaneously indicating that the model prediction has a high degree of accuracy.

Predicted uniaxial tensile stress–strain curves at five different stretching speeds are shown in Figure 9, indicating that the modified model accurately forecasts the stress–strain relationship trends of rubber materials at different strain rates without the need to refit parameters. It is also evident that varying stretching speeds significantly impact the material’s stress–strain curves, with higher speeds resulting in steeper slopes, which further confirms the inability of the Yeoh model to simulate behaviors in the small strain range accurately.

The figure also reflects the influence of the strain rate on the hyperelastic properties of the materials within the small strain region. Rubber materials display similar nonlinear behaviors at different strain rates in the forecasted stretching results. The material’s stiffness decreases with increasing applied strain, while at a constant strain, the stiffness increases with an increasing strain rate. Therefore, the initial stiffness is closely related to the strain rate—the higher the strain rate, the greater the material stiffness (i.e., modulus). This phenomenon is consistent with the conclusions of Sanghyeub Kim et al. [40], who found that different strain rates lead to variations in the entanglement and disentanglement of chains in uncured rubber. In vulcanized rubber, a network of chain entanglements also exists, and the contribution of this network to the modulus significantly increases under high-speed strain [41].

### 3.7. The Impact of the Coagulation Method on the Modified Model Parameters of Rubber in Small Strain Regions

The five rubber samples were stretched to 100% strain at different tensile speeds of 300 mm/min and 500 mm/min. The tensile speed of 300 mm/min was selected as the reference speed, and the model parameters were calculated according to the Formula (9). The changes in the model parameters (C_10_) before and after the modification are shown in Table 10.

After the modification, the model parameter C_10_ increased for all five samples, and the differences between them were amplified. However, there was no change in the order of their values, indicating that the natural coagulant is still the largest and the acid coagulant is the smallest in terms of modulus, which is in agreement with the results of the study before the model modification.

## 4. Conclusions

Based on tensile experiments and the Yeoh model, the effects of natural coagulation, enzyme coagulation, acid coagulation, microbial coagulation, and enzyme-assisted microbial coagulation modes on the superelastic ontological relationships of natural rubber were analyzed in detail. Given the limitations of the Yeoh model in describing mechanical behavior in the low-strain region, a strain rate-related factor was introduced, and a more accurate hyperelastic constitutive relationship of natural rubber was established. The results demonstrate that the Yeoh model effectively characterizes the mechanical behavior of natural rubber carbon black composites in the moderate and large strain regions. Additionally, the material characteristics reflected by the model parameters align with those observed in cyclic tensile tests. The performance of rubber under small, medium, and large deformations is different for different coagulation methods. The modulus of natural coagulation rubber is the highest in small deformations, the modulus of enzyme coagulation rubber is the highest in medium deformations, and the modulus of enzyme-assisted microbial coagulation rubber is the highest in large deformations, and thus the appropriate coagulation method can be selected according to the degree of deformation of the rubber required by application scenarios in the practical application. Following the incorporation of a strain rate-related factor, the modified Yeoh model accurately predicts stress–strain curves across different tensile speeds within the low-strain range.

## Figures and Tables

**Figure 1 polymers-16-03601-f001:**
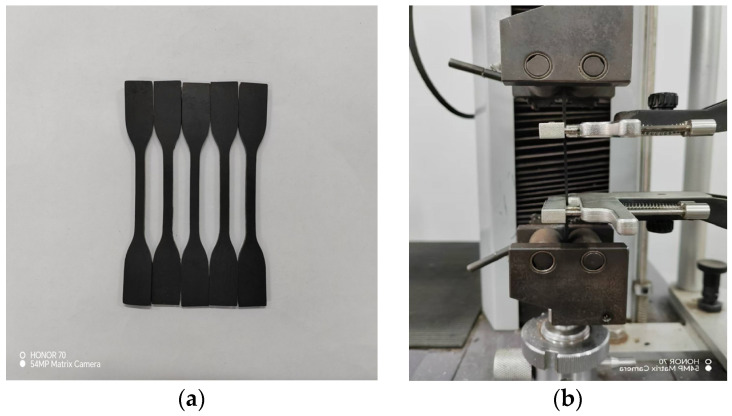
Tensile samples and instrument: (**a**): samples; (**b**): instrument.

**Figure 2 polymers-16-03601-f002:**
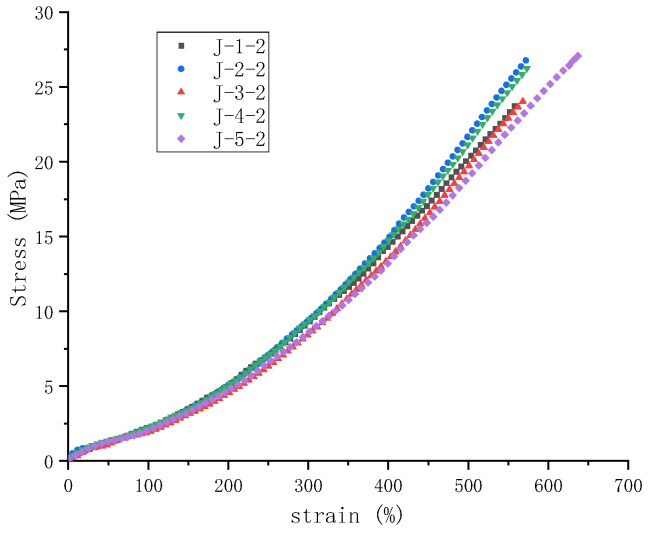
Stress–strain curves of natural rubber materials under different coagulation methods.

**Figure 3 polymers-16-03601-f003:**
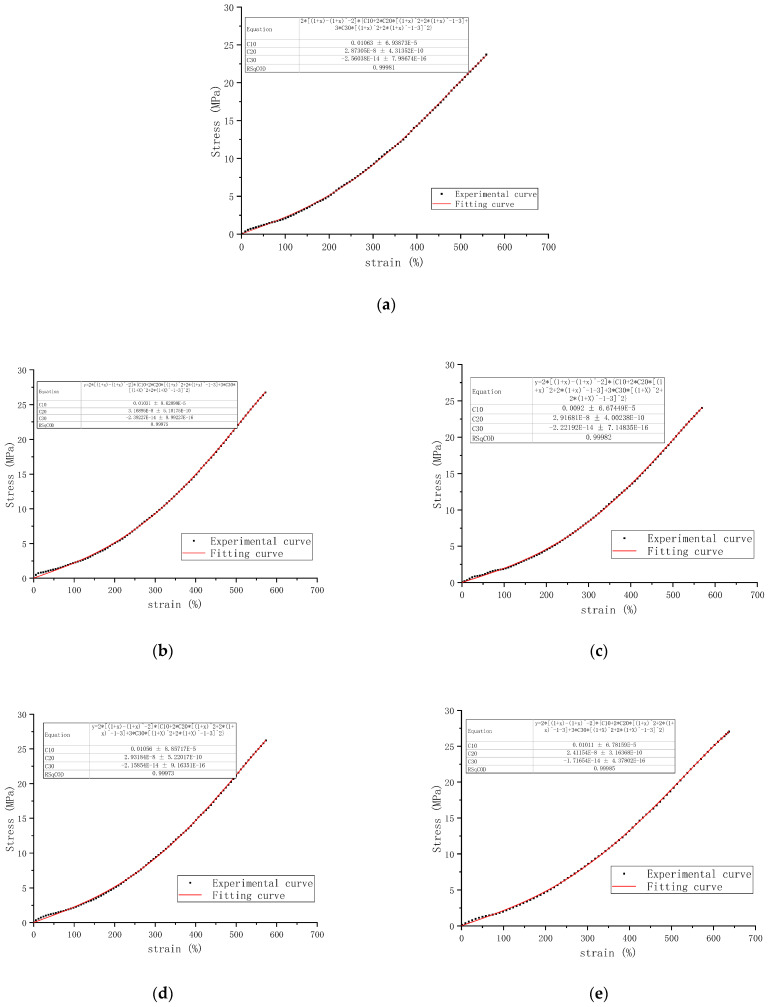
Yeoh model’s fitting curve of rubber samples: (**a**) J-1-2; (**b**) J-2-2; (**c**) J-3-2; (**d**) J-4-2; and (**e**) J-5-2.

**Figure 4 polymers-16-03601-f004:**
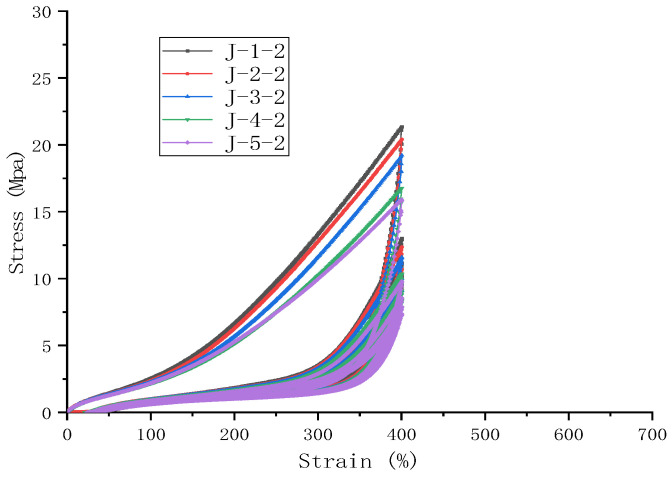
The effect of the coagulation method on the cyclic stretching curves.

**Figure 5 polymers-16-03601-f005:**
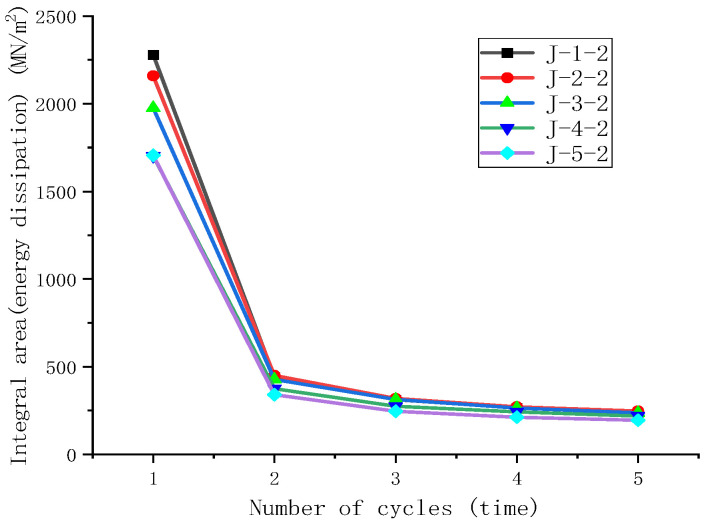
The effect of the coagulation method on energy dissipation.

**Figure 6 polymers-16-03601-f006:**
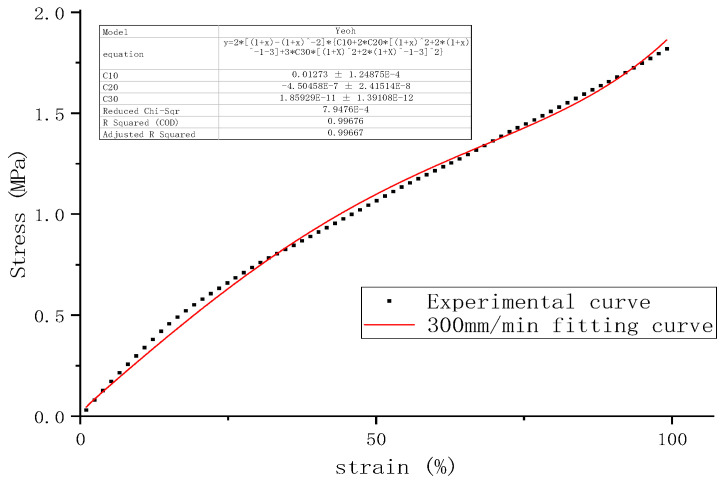
The actual stress–strain curve and the fitting curve at the reference speed of 300 mm/min.

**Figure 7 polymers-16-03601-f007:**
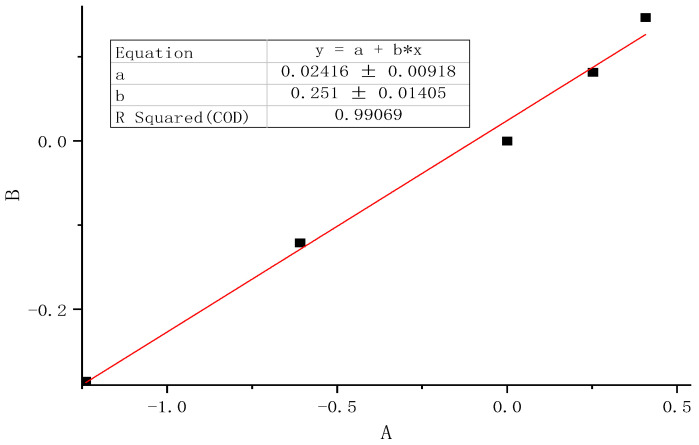
Linear fitting of the strain rate-related parameter μ.

**Figure 8 polymers-16-03601-f008:**
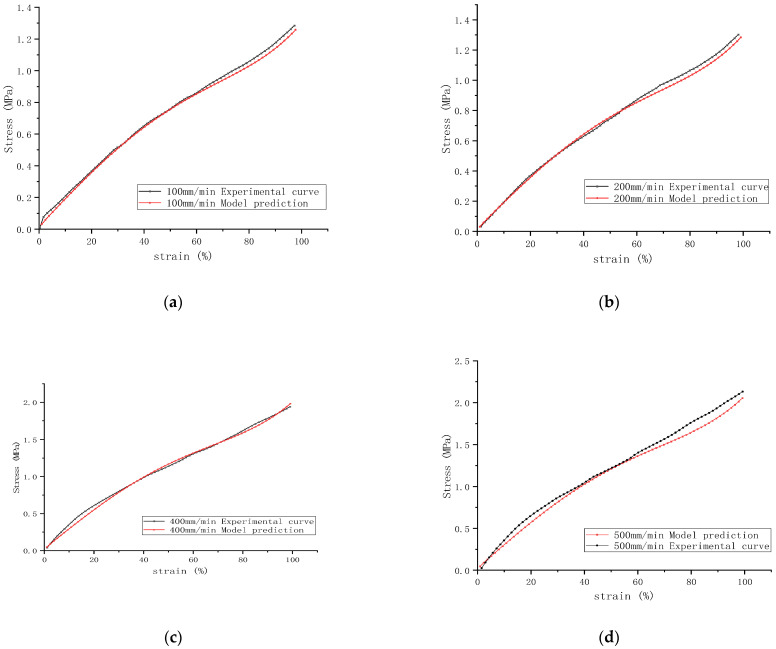
A comparison between the curve calculated by the modified model and the actual curve at different speeds: (**a**) 100 mm/min; (**b**) 200 mm/min; (**c**) 400 mm/min; and (**d**) 500 mm/min.

**Figure 9 polymers-16-03601-f009:**
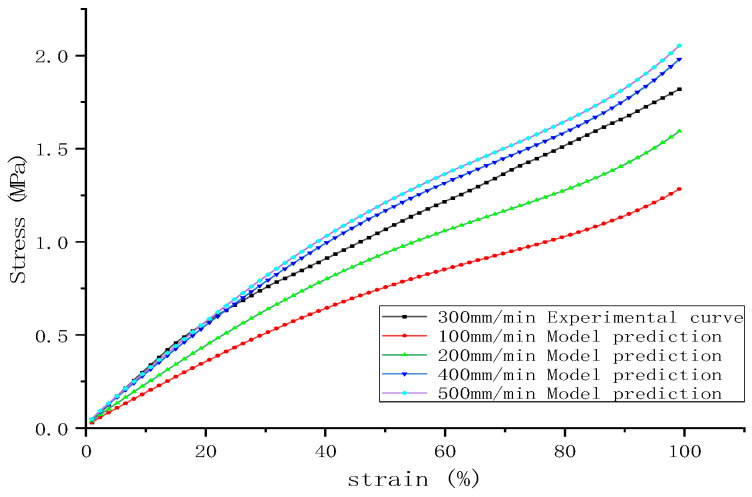
Predicted uniaxial tensile stress–strain curves.

**Table 1 polymers-16-03601-t001:** Sample preparation.

Samples Number	Coagulation Methods	Type of Coagulant	Dosage of Coagulant	Note
J-1-2	Natural coagulation	—	—	
J-2-2	Enzymatic coagulation	Alkaline protease	Weight of dry rubber × 0.09%	
J-3-2	Acid coagulation	Methanoic acid	Weight of dry rubber × 0.35%	
J-4-2	Microbial coagulation	Microbial coagulation liquid	Weight of latex × 10%	
J-5-2	Enzyme-assisted microbial coagulation	a. Alkaline protease	Weight of dry rubber × 0.05%	“a” was added a first and mixed well.
		b. Microbial coagulation liquid	Weight of latex × 10%	After 20 min, “b” was added

**Table 2 polymers-16-03601-t002:** Formulation of NR compounds.

Chemical	Quantity (phr)
Natural rubber	100.00
Stearic acid	2.00
N-tert-butyl-2-benzothiazole sulfenamide (TBBS)	0.70
Zinc oxide	5.00
Sulfur	2.25
Oil furnace Black	35.00

**Table 3 polymers-16-03601-t003:** Tensile property of natural rubber materials by different coagulation methods.

Samples Number	Tensile Strength/MPa	Elongation at Break/%	Modulus at 100%/MPa	Modulus at 300%/MPa	Modulus at 500%/MPa
J-1-2	23.72	558.10	2.11	9.18	20.23
J-2-2	26.76	572.12	2.24	9.30	21.71
J-3-2	24.02	568.32	1.92	8.48	19.67
J-4-2	26.22	573.26	2.18	9.37	20.18
J-5-2	27.06	637.60	2.10	8.44	19.01

**Table 4 polymers-16-03601-t004:** Variance analysis of group.

Group	Number of Observations	Summation	On Average	Variance (Statistics)
J-1-2	5	0.179503	0.035901	0.002863
J-2-2	5	0.487834	0.097567	0.003336
J-3-2	5	0	0	0
J-4-2	5	5.36658	1.073316	0.002903
J-5-2	5	5.303944	1.060789	0.005585

**Table 5 polymers-16-03601-t005:** Variance analysis of intergroup.

Source of Variation	SS	df	MS	F	*p*-Value	F Crit
intergroup	6.298557	4	1.574639	536.0519	4.95 × 10^−20^	2.866081

**Table 6 polymers-16-03601-t006:** Constants of Yeoh model.

Samples Number	C_10_	C_20_	C_30_
J-1-2	0.01063 ± 6.93873 × 10^−5^	2.87305 × 10^−8^ ± 4.31352 × 10^−10^	−2.56038 × 10^−14^ ± 7.98674 × 10^−16^
J-2-2	0.01031 ± 8.62098 × 10^−5^	3.16895 × 10^−8^ ± 5.10175 × 10^−10^	−2.39227 × 10^−14^ ± 8.99227 × 10^−16^
J-3-2	0.0092 ± 6.67449 × 10^−5^	2.91681 × 10^−8^ ± 4.00238 × 10^−10^	−2.22192 × 10^−14^ ± 7.14835 × 10^−16^
J-4-2	0.01056 ± 8.85717 × 10^−5^	2.93184 × 10^−8^ ± 5.22017 × 10^−10^	−2.15854 × 10^−14^ ± 9.16351 × 10^−16^
J-5-2	0.01011 ± 6.78159 × 10^−5^	2.41154 × 10^−8^ ± 3.16368 × 10^−10^	−1.71654 × 10^−14^ ± 4.37802 × 10^−16^

**Table 7 polymers-16-03601-t007:** The differences between the theoretical and actual strain rates at different speeds.

Stretching Speeds (mm/min)	100	200	300	400	500
$V_{\varepsilon }^{f}$ theoretical strain rate (/s)	0.0667	0.1333	0.2	0.2667	0.3333
Strain time (s)	31.27	16.11	9.05	7.03	6.02
$V_{\varepsilon }^{f}$ actual strain rates (/s)	0.032	0.0621	0.1105	0.1422	0.1661

**Table 8 polymers-16-03601-t008:** The differences between the theoretical and actual strain rates at different speeds.

Stretching Speeds (mm/min)	$V_{\varepsilon }^{f}$ Actual Strain Rates (/s)	$C_{10}^{f}$	ln($V_{\varepsilon }^{f}$$/V_{\varepsilon_{0}}$)	$C_{10}^{f}$$/C_{10}^{\varepsilon_{0}}$ − 1
100	0.032	0.0091	−1.239279618	−0.285153181
200	0.0621	0.01119	−0.609005679	−0.120974077
300	0.1105 ($V_{\varepsilon_{0}}$)	0.01273 ($C_{10}^{\varepsilon_{0}}$)	0	0
400	0.1422	0.01377	0.252218996	0.081696779
500	0.1661	0.0146	0.407574496	0.146897093

**Table 9 polymers-16-03601-t009:** Accuracy of model predictions.

Stretching Speeds (mm/min)	R^2^ (COD)
100 (actual test)	0.99932
100 (model prediction)	1
200 (actual test)	0.99921
200 (model prediction)	1
400 (actual test)	0.99624
400 (model prediction)	1
500 (actual test)	0.99657
500 (model prediction)	1

**Table 10 polymers-16-03601-t010:** The model parameters (C_10_) before and after the modification.

Samples Number	$V_{\varepsilon }^{f}$ Actual Strain Rates (/s)		Yeoh Model C_10_		Yeoh Modified Model C_10_
	300 mm/min	500 mm/min	300 mm/min	500 mm/min	500 mm/min
J-1-2	0.1105	0.1661	0.01615	0.0146	0.017802
J-2-2	0.0994	0.1661	0.01427	0.01353	0.016109
J-3-2	0.0994	0.1661	0.01263	0.01237	0.014258
J-4-2	0.0903	0.1658	0.01438	0.01455	0.016233
J-5-2	0.0994	0.1661	0.01273	0.01288	0.014672

## Data Availability

Data are contained within the article.

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
