# Peer review of "The Influence of Fresh Latex Coagulation on the Parameter Characteristics of the Yeoh Hyperelastic Constitutive Model for Natural Rubber"

_polymers, 2024, doi:10.3390/polym16243601_

Round 1

Reviewer 1 Report

Comments and Suggestions for Authors

My comments are in the attached document

Comments on the Quality of English Language

English needs improvement. 

Author Response

Response to Reviewer 1 Comments

1. Summary

Dear Reviewer:

Thank you very much for the comments concerning our manuscript entitled “Influence of Fresh Latex Coagulation on the Parameter Characteristics of the Yeoh Hyperelastic Constitutive Model for Natural Rubber (Manuscript Number polymers-3362404)”. These comments are valuable and helpful for revising and improving our paper. We have made extensive corrections to our previous draft. These changes are highlighted in red in the revised manuscript. The detailed corrections are listed below.

2. Questions for General Evaluation

Reviewer’s Evaluation

Response and Revisions

Does the introduction provide sufficient background and include all relevant references?

Can be improved

Thank you for this comment. We have improved.

Is the research design appropriate?

Yes

Thank you for this comment.

Are the methods adequately described?

Must be improved

Thank you for this comment. We have improved.

Are the results clearly presented?

Must be improved

Thank you for this comment. We have improved.

Are the conclusions supported by the results?

Yes

Thank you for this comment.

3. Point-by-point response to Comments and Suggestions for Authors

Comments 1: This work studies the modiffcation of the Yeoh model to predict mechanical properties of NR obtained from different types of latex (obtained with different coagulation methods). The work is well performed and structured and the analysis of the results is well done. However, the novelty and the importance of this study is not so clear.

Response 1: Thank you for this comment. We have carefully modified and changed contents such as the “Abstract,”” Introduction,” and “Conclusions “ so that the novelty and the importance of this study are clearer than before. These changes are highlighted in red on pages 1, 2, and 15 of the revised manuscript.

Comments 2: Some comments and recommendations: Abstract. --The abstract is too long and dense, it would be recommended to summarize the main findings of the study in a more simple and clear way.

Response 2: Thank you for pointing this out. we have revised the abstract. This change is highlighted in red on page 1, in lines 15-29 of the revised manuscript.

Comments 3: Some comments and recommendations: Introduction. --The introduction has a nice overview about Natural Rubber. However more information about the Yeoh model, why is it important and the novelty of the introduction of a strain rate factor should be included. It is not clear what this model and not another was chosen for this study.

Response 3: Thank you for pointing this out. We have added more information about the Yeoh model. The Yeoh model is more suitable for simulating the significant deformation behavior of carbon black-filled natural rubber, the most commonly used filler in the natural rubber industry. Choosing the Yeoh model is more beneficial for future research. These changes are highlighted in red on page 2, lines 74-96 of the revised manuscript.

Comments 4: Some comments and recommendations: Materials and methods. --This section is a bit unorganized. A better structure would help to make reading easier. A more detailed description of the preparation of the samples with the different coagulation methods should be included.

Response 4: Thank you for pointing this out. We have revised “Materials and methods”. A more detailed description of the preparation of the samples with the different coagulation methods is already included. These changes are highlighted in red on page 3, in lines 108-181 of the revised manuscript.

Comments 5: Some comments and recommendations: Results and discussion. --Figures like stress-strain curves (Figure 1, Figure 7) should use consistent formatting (e.g., axis labels, units) for easier comparison. Also in some of these graphs the data is barely readable. It should be improved. Furthermore, the numerical results (values for tensile strength, modulus and elongation at break) should be presented in a table. Also standard deviations should be added to be able to determine the reliability of the test and if there is signiffcant difference between the results of each coagulation method.

Response 5: Thank you for pointing this out. We have modified the Figures (Figures 2-8), ensuring they maintain consistent formatting and are readable. Furthermore, we have added the variance analysis of the tensile property of NR by different coagulation methods (Table 3-4). These changes are highlighted in red on page 6, lines 206-217 of the revised manuscript.

4. Response to Comments on the Quality of English Language

Point 1: Sentences could be shortened to make the reading experience easier.

Response 1: Thank you for pointing this out. We have made extensive corrections to our previous draft, which we hope will make reading easier. These can be found in the revised manuscript.

Point 2: In general, English should be improved.

Response 2: Many thanks for the comment. We have carefully revised the manuscript according to the reviewers’ comments and have also re-scrutinized it to improve the English with a language polishing service.

Point 3: Inconsistent use of technical terms (e.g., "strain rate parameter" vs. "strain-raterelated factor"). It would be better to use the same terminology for the whole manuscript.

Response 3: Many thanks for the comment. We have consistently changed the technical terms for the whole manuscript, such as “strain rate-related”. However, the “strain rate-related factor” is not the "strain rate parameter." They have different physical meanings.

Point 4: Same for ffgures, in some cases is written “as seen in Figure 5…” and in others

“Refer to Fig.3 for …”. Again consistency is missing, it should be the same for the

whole manuscript.

Response 4: Many thanks for the comment. We have consistently changed the expression of the citation for the charts and graphs throughout the manuscript. These changes are highlighted in red in lines 121, 146, 156, 158, 207, 211, 223, 276, 277, 313, 353, 365, and 396 of the revised manuscript.

5. Additional clarifications

Thank you very much for the comments and suggestions. I hope the re-submission manuscript will be acceptable for publication in the journal. If there are any problems or questions about our paper, please do not hesitate to let us know.

Reviewer 2 Report

Comments and Suggestions for Authors

1.      The abstract lacks details in regards to effectively illustrate how this research contributes to the advancement of natural rubber processing or material science, lacking a clear and compelling statement of the study's contributions.

2.      The term "properties" in the keywords is too general and should be substituted with a more specific word.

3.      Line 66: Ng should be corrected to Ng et al.

4.      The main objectives and innovation of the paper must be written in a clearer and more concise way at the end of the introduction section.

5.      The novelty of this work needs to be elaborated more. Why this study is so important? How it differentiates from the other methods.

6.      The experimental section lacks rigor and omits critical information necessary for reproducing the proposed experiments. Additionally, details such as sample sizes, specific procedures, and calibration methods should be included to enhance reproducibility and ensure clarity in the experimental design.

7.      Include images of the actual samples along with their dimensions, as well as the setup used for the tensile test.

8.      Provide more details about the compounded rubber preparation formula No. 3 and the references regarding the use of a vulcanization time of 30 minutes at 143°C?

9.      Ensure that appropriate references should be included to all mathematical expressions from 1 to 10.

10.  The equations contain several symbols that haven't been defined.

11.  In Figure 1, discuss the reasons for the nearly identical data observed for J-4-2 and J-5-2, as well as for J-1-2 and J-3-2.

12.  The article mentions using the Yeoh model for fitting the stress-strain data but fails to justify why this particular model was selected over others. Without a rationale for the model choice, the validity of the results is called into question.

13.  While Table 2 presents the fitted parameters, it does not describe the methodology used for fitting. Details regarding the optimization process or the assumptions underlying the model are necessary for evaluating the robustness of the results.

14.  The claims regarding the order of rubber hardness and softening based on coagulation methods are presented without sufficient context or references support.

15.  In Figures 3 and 4, the relationship between energy dissipation and material softness/hardness is stated but could be supported with additional data or references. How do these findings align with existing literature?

16.  Figures 3 and 4 were not referenced in the text.

17.  The process for calculating the theoretical stress-strain curve data is not adequately explained.

18.  There is no mention of any statistical tests or validations performed on the model predictions. Including such analyses would strengthen the argument and provide a more robust foundation for the claims made.

19.  Conclusion should clearly mention the findings, most preferably quantitatively. It is better if the conclusion is represented point-wise.

2

Author Response

Response to Reviewer 2 Comments

1. Summary

Dear Reviewer:

Thank you very much for the comments concerning our manuscript entitled “Influence of Fresh Latex Coagulation on the Parameter Characteristics of the Yeoh Hyperelastic Constitutive Model for Natural Rubber (Manuscript Number polymers-3362404)”. These comments are valuable and helpful for revising and improving our paper. We have made extensive corrections to our previous draft. These changes are highlighted in red in the revised manuscript. the detailed corrections are listed below.

2. Questions for General Evaluation

Reviewer’s Evaluation

Response and Revisions

Does the introduction provide sufficient background and include all relevant references?

Must be improved

Thank you for this comment. We have improved.

Is the research design appropriate?

Must be improved

Thank you for this comment. We have improved.

Are the methods adequately described?

Must be improved

Thank you for this comment. We have improved.

Are the results clearly presented?

Must be improved

Thank you for this comment. We have improved.

Are the conclusions supported by the results?

Must be improved

Thank you for this comment. We have improved.

3. Point-by-point response to Comments and Suggestions for Authors

Comments 1: The abstract lacks details in regards to effectively illustrate how this research contributes to the advancement of natural rubber processing or material science, lacking a clear and compelling statement of the study's contributions.

Response 1: Thank you for this comment. We have carefully modified and changed the “Abstract”. These changes are highlighted in red on page 1, in lines 15-29 of the revised manuscript.

Comments 2: The term "properties" in the keywords is too general and should be substituted with a more specific word.

Response 2: Thank you for this comment. We have changed the” properties” to “Ng et al. “on page 1, in line 30 of the revised manuscript.

Comments 3: Line 66: Ng should be corrected to Ng et al.

Response 3: Thank you for this comment. We have changed the” Ng” to “Ng et al. “on page 2, in lines 97 and 63 of the revised manuscript.

Comments 4: The main objectives and innovation of the paper must be written in a clearer and more concise way at the end of the introduction section.

Response 4: Thank you for this comment. We have carefully modified and changed the end of the introduction section. These changes are highlighted in red on page 2, in lines 97 and 106 of the revised manuscript.

Comments 5: The novelty of this work needs to be elaborated more. Why this study is so important? How it differentiates from the other methods.

Response 5: Thank you for this comment. We have carefully modified and changed contents such as the “Abstract,”” Introduction,” and “Conclusions “ so that the novelty and the importance of this study are clearer than before. These changes are highlighted in red on pages 1, 2, and 15 of the revised manuscript.

Comments 6: The experimental section lacks rigor and omits critical information necessary for reproducing the proposed experiments. Additionally, details such as sample sizes, specific procedures, and calibration methods should be included to enhance reproducibility and ensure clarity in the experimental design

Response 6: Thank you for this comment. We have revised “Materials and methods.” A more detailed description of the materials and methods is already included. These changes are highlighted in red on page 3, lines 108-181 of the revised manuscript.

Comments 7:  Include images of the actual samples along with their dimensions, as well as the setup used for the tensile test

Response 7: Thank you for this comment. We have revised “Materials and methods.” A more detailed description of the preparation methods is already included. We also gave the images. These changes are highlighted in red on page 3, lines 108-181 of the revised manuscript.

Comments 8:  Provide more details about the compounded rubber preparation formula No. 3 and the references regarding the use of a vulcanization time of 30 minutes at 143°C?

Response 8: Thank you for this comment. We have revised “Materials and methods.” A more detailed description of the preparation methods is already included. These changes are highlighted in red on page 3, lines 108-181 of the revised manuscript.

Comments 9:  Ensure that appropriate references should be included to all mathematical expressions from 1 to 10.

Response 9: Thank you for this comment. We have added the references of the mathematical expressions from 1 to 10. These changes are highlighted in red in lines 169, 172, 173, 177, 301, and 326 of the revised manuscript.

Comments 10: The equations contain several symbols that haven't been defined.

Response 10: Thank you for this comment. We have added the definitions of the symbols in equations. These changes are highlighted in red on page 5, in lines 170 and 175-176 of the revised manuscript.

Comments 11:  In Figure 1, discuss the reasons for the nearly identical data observed for J-4-2 and J-2-2, as well as for J-1-2 and J-3-2.

Response 11: Thank you for this comment. We have added a discussion of the reasons for the nearly identical data observed for J-2-2 and J-3-2, as well as for J-1-2 and J-3-2. These changes are highlighted in red on page 7, lines 233-240 of the revised manuscript.

Comments 12:  The article mentions using the Yeoh model for fitting the stress-strain data but fails to justify why this particular model was selected over others. Without a rationale for the model choice, the validity of the results is called into question.

Response 12: Thank you for this comment. We have checked the existing literature. The result shows that the Yeoh model is more suitable for simulating the significant deformation behavior of carbon black-filled natural rubber, the most commonly used filler in the natural rubber industry. Choosing the Yeoh model is more beneficial for future research. These are highlighted in red on page 2, lines 74-96 of the revised manuscript.

Comments 13: While Table 2 presents the fitted parameters, it does not describe the methodology used for fitting. Details regarding the optimization process or the assumptions underlying the model are necessary for evaluating the robustness of the results.

Response 13: Thank you for this comment. We have added the methodology used for fitting. the strain and stress data obtained from the tests were nonlinearly curve-fitted by the above equations in the software Origin 2021 to obtain the parameters C10, C20, and C30 of the Yeoh model. These changes are highlighted in red on page 5, lines 179-181 of the revised manuscript.

Comments 14:  The claims regarding the order of rubber hardness and softening based on coagulation methods are presented without sufficient context or references support.

Response 14: Thank you for this comment. We have referenced [38] and [39] to support the claims regarding the order of rubber hardness and softening based on coagulation methods. These changes are highlighted in red on page 17, lines 531-532 of the revised manuscript.

Comments 15:  In Figures 3 and 4, the relationship between energy dissipation and material softness/hardness is stated but could be supported with additional data or references. How do these findings align with existing literature?

Response 15: Thank you for this comment. We have checked the existing literature, and there are a few references to the Yeoh Model in the study of the NR. The relationship between energy dissipation and material softness/hardness is also a new point of view. We hope to go on to do more work on it.

Comments 16: Figures 3 and 4 were not referenced in the text.

Response 16: Thank you for this comment. We have referenced those two Figures. These changes are highlighted in red on page 11, lines 274 and 227 of the revised manuscript.

Comments 17: The process for calculating the theoretical stress-strain curve data is not adequately explained.

Response 17: Thank you for this comment. We have added the process for calculating the theoretical stress-strain curve data. The theoretical strain data for each speed can be calculated by substituting the values of C10, C20, C30 at 300mm/min tensile speed and “ln( / )” at that speeds under 300mm/min tensile speed in Table 7 into Equation 8. These changes are highlighted in red on page 13, lines 349-352 of the revised manuscript. And also gave the process for calculating the Strain rate. These changes are highlighted in red on page 11, lines 313-318 of the revised manuscript.

Comments 18:  There is no mention of any statistical tests or validations performed on the model predictions. Including such analyses would strengthen the argument and provide a more robust foundation for the claims made.

Response 18: Thank you for this comment. We have added the “Accuracy of model predictions. (Table 8.)” These changes are highlighted in red on page 14, lines 360-369 of the revised manuscript.

Comments 19: Conclusion should clearly mention the findings, most preferably quantitatively. It is better if the conclusion is represented point-wise.

Response 19: Thank you for this comment. We have carefully modified and changed the “Conclusions.” These changes are highlighted in red on page 15, lines 405-423 of the revised manuscript.

4. Response to Comments on the Quality of English Language

There are no Comments. We have carefully revised the manuscript and have also re-scrutinized it to improve the English with a language polishing service.

5. Additional clarifications

Thank you very much for the comments and suggestions. I hope the re-submission manuscript will be acceptable for publication in the journal. If there are any problems or questions about our paper, please do not hesitate to let us know.

Round 2

Reviewer 1 Report

Comments and Suggestions for Authors

Suggestions were implemented accordingly

Reviewer 2 Report

Comments and Suggestions for Authors

The authors have effectively addressed the majority of my comments and have made significant improvements to the manuscript. Their revisions enhance the overall clarity and depth of the work, demonstrating a thoughtful engagement with the feedback provided.